# Graphene-Based Flexible Electrode for Electrocardiogram Signal Monitoring

Tian-Rui Cui [1,2,†] , Ding Li [1,2] , Xiao-Rui Huang [1,2], An-Zhi Yan [1,2], Yu Dong [1,2], Jian-Dong Xu [1,2], Yi-Zhe Guo [1,2], Yu Wang [1,2], Zhi-Kang Chen [1,2], Wan-Cheng Shao [1,2], Ze-Yi Tang [1,2], He Tian [1,2,*,†], Yi Yang [1,2,*] and Tian-Ling Ren [1,2,3,*]

1   School of Integrated Circuit, Tsinghua University, Beijing 100084, China
2   Beijing National Research Center for Information Science and Technology (BNRist), Tsinghua University, Beijing 100084, China; ctr19@mails.tsinghua.edu.cn (T.-R.C.); lidingli0813@163.com (D.L.); hxr19@mails.tsinghua.edu.cn (X.-R.H.); yaz21@mails.tsinghua.edu.cn (A.-Z.Y.); dongyu21@mails.tsinghua.edu.cn (Y.D.); xjd18@mails.tsinghua.edu.cn (J.-D.X.); guoyz20@mails.tsinghua.edu.cn (Y.-Z.G.); wangyu20@mails.tsinghua.edu.cn (Y.W.); 17709284348@mail.nwpu.edu.cn (Z.-K.C.); wancshao218@163.com (W.-C.S.); tzy18874264030@163.com (Z.-Y.T.)
3   Center for Flexible Electronics Technology, Tsinghua University, Beijing 100084, China
*   Correspondence: tianhe88@tsinghua.edu.cn (H.T.); yiyang@tsinghua.edu.cn (Y.Y.); rentl@tsinghua.edu.cn (T.-L.R.)
†   These authors contributed equally to this work.

**Abstract:** With the rapidly aging society and increased concern for personal cardiovascular health, novel, flexible electrodes suitable for electrocardiogram (ECG) signal monitoring are in demand. Based on the excellent electrical and mechanical properties of graphene and the rapid development of graphene device fabrication technologies, graphene-based ECG electrodes have recently attracted much attention, and many flexible graphene electrodes with excellent performance have been developed. To understand the current research progress of graphene-based ECG electrodes and help researchers clarify current development conditions and directions, we systematically review the recent advances in graphene-based flexible ECG electrodes. Graphene electrodes are classified as bionic, fabric-based, biodegradable, laser-induced/scribed, modified-graphene, sponge-like, invasive, etc., based on their design concept, structural characteristics, preparation methods, and material properties. Moreover, some categories are further divided into dry or wet electrodes. Then, their performance, including electrode–skin impedance, signal-to-noise ratio, skin compatibility, and stability, is analyzed. Finally, we discuss possible development directions of graphene ECG electrodes and share our views.

**Keywords:** graphene electrode; ECG; flexible devices; wearable electronics

## 1. Introduction

Cardiovascular diseases, including ischemic heart disease and stroke, are the top global causes of death in the world [1]. Although many different factors contribute to cardiovascular diseases, both aging and unbalanced medical resources lower the threshold for the manifestation of cardiovascular diseases [2]. Populations are becoming older in most developed regions and poor medical conditions in some developing regions [3] are contributing to a further increase in the incidence of cardiovascular disease, making it a global, universal problem. In the face of one of humanity's leading health threats and people's increasing concern for cardiovascular health, long-term electrocardiogram (ECG) signal monitoring is particularly important for early diagnosis and prevention of cardiovascular diseases [4]. With this comes the need for flexible electrodes with high performance and suitable for long-term ECG monitoring.

The broadly used commercial Ag/AgCl gel electrode plays an important role in long-term ECG monitoring as it is low-cost, has achieved large-scale manufacturing, and

conforms well to most people's skin [5]. However, its shortcomings have also been exposed over time [6,7]. For example, the gel in Ag/AgCl electrode is airtight and contains ions, which leads to the accumulation of sweat and might cause irritation during long-term ECG monitoring. Besides, dehydration of the gel during long-term wearing reduces its flexibility and increases electrode–skin contact impedance, which reduces comfort and affects the quality of ECG monitoring.

The rapid development of wearable electronics [8–10] has advanced ECG electrodes while creating the development conditions for both high-quality and long-term monitoring of ECG signals [11,12]. For high-quality monitoring, the electrode should have low skin–electrode contact impedance [13] and a high signal-to-noise ratio [14]. For long-term monitoring, the electrode should be flexible [15], biocompatible [16], stable [17], and comfortable to wear [18].

As a two-dimensional material with excellent properties, graphene has many physical and chemical advantages, including high mobility up to 10,000 $cm^{-2} \cdot s^{-1}$ at room temperature, high thermal conductivity over 5000 $W \cdot mK^{-1}$ for a single-layer sheet, good intrinsic mechanical properties with 42 $N \cdot m^{-1}$ breaking strength and 1.0 TPa Young's modulus (making it one of the strongest materials ever tested), and good chemical inertness due to its covalent bond structure [19–21]. These excellent properties enable graphene to have a bright performance in biomedical applications in recent years, especially in wearable biomedical electronics [22,23]. As one of the most prominent biomedical applications of graphene, when used as flexible electrode material graphene can contact human skin with high conformability [24,25], which is beneficial to reduce contact impedance between the electrode and skin [26], improve the signal-to-noise ratio [27], and ensure high-quality signal acquisition [28]. Besides, graphene shows stability and biocompatibility for on-skin applications [29]. Combining the excellent properties of graphene, it is an ideal material for flexible ECG electrodes for long-term, high-quality ECG signal monitoring.

When used for ECG signal monitoring, graphene electrodes can obtain ECG signals in different manners, including skin surface contact, implantable contact, and indirect contact [8]. These monitoring methods all obtain the ECG signals by recording the small voltage changes produced by the depolarization and repolarization of the human heart [11]. Besides, in the case of graphene electrodes, the process of monitoring ECG signals involves converting the ionic current generated by the heart into an electric current in the electrode. Graphene ECG electrodes act as a transducer during ECG monitoring [12]. Since the ECG signal is a small signal with long-term dynamic changes, a graphene ECG electrode is required to achieve long-term ECG monitoring with high quality, high precision, and high stability.

In this review, we introduce the recent advances in graphene-based flexible ECG electrodes. The first part introduces currently-proposed graphene ECG electrodes, which are generally divided into seven categories: bionic, fabric-based, biodegradable, laser-induced/scribed, modified-graphene, sponge-like, and invasive based on their design philosophy, structural characteristics, preparation methods, material properties, etc. Besides, some categories are further divided into dry and wet electrodes. Then, their performance, including electrode–skin impedance, signal-to-noise ratio, skin compatibility, and stability are further analyzed and discussed. The second part discusses the possible development directions of graphene ECG electrodes, with the help of some typical research progress on wearable flexible graphene-based ECG electrodes and systems. Finally, we give the future development prospects of graphene ECG electrodes from our perspective and conclude.

## 2. Research on Graphene-Based Flexible ECG Electrodes

Human skin is an irregular and flexible surface. Thus, compared to flexible electrodes, the use of rigid electrodes may cause many problems [30,31]. For instance: (i) Rigid electrodes cannot adhere to the skin with high compatibility, which means the electrodes cannot maintain good electrical contact with the skin, resulting in high electrode–skin impedance. (ii) When people are going about their daily activities, rigid electrodes are easy to move rela-

tive to the skin and will introduce movement noise during long-term monitoring. (iii) Rigid electrodes are uncomfortable and may cause skin damage during long-term wearing. Based on the disadvantages of rigid ECG electrodes, many flexible ECG electrodes have been developed [32,33]. Flexible ECG electrodes are a kind of ECG electrode that can be uniformly attached to the human skin (a few of which are invasive) for ECG monitoring. With high flexibility, flexible ECG electrodes can follow skin movement and reduce motion noise interference. Besides, good contact between the flexible electrode and skin can reduce electrode–skin contact impedance and ensure the quality of the collected ECG signal. Moreover, except for a few for implantable monitoring devices, flexible ECG electrodes do not damage the skin, which improves comfort and is suitable for long-term wear.

When combined with the excellent properties of graphene, a variety of graphene-based flexible ECG electrodes have been developed. According to their different design methods, structures, materials, etc., we divide them into bionic, fabric-based, biodegradable, laser-induced/scribed, modified-graphene, sponge-like, invasive, and other flexible electrodes, and introduce their respective characteristics.

## 2.1. Bionic Graphene ECG Electrodes

Inspired by nature, many bionic flexible devices for different applications have been developed, such as electronic skin [32], wearable sensors [33], and flexible electrodes [34]. By mimicking human and biological structures, a variety of ECG electrodes with excellent performance, including better skin adhesion, low skin contact impedance, and high stability, have been proposed [35–39]. In this section, we introduce several typical biomimetic graphene ECG electrodes that have been developed in recent years and discuss their performance characteristics.

### 2.1.1. Gecko-Inspired ECG Electrode

Conventional dry ECG electrodes usually require auxiliary adhesion in the absence of their adhesive layer, which means contact between electrode and skin is not tight enough, resulting in high electrode–skin impedance. Besides, the auxiliary adhesion may decrease in viscosity over time, leading to the deterioration of ECG signal quality, which is not suitable for long-term ECG monitoring. Gecko-inspired structures consisting of high-aspect-ratio nanopillars have been proven to be able to repeatedly adhere and detach from human skin with the help of van der Waals interactions [35]. By combining high conductance, high flexibility, and excellent adhesion properties, some gecko-inspired graphene ECG electrodes have been developed recently. Kim et al. proposed a concept of conductive dry adhesives (CDA) [36] by using graphene, carbon nanotubes, and other higher dimensional carbon materials to form the conductive network in the polydimethylsiloxane (PDMS) elastomeric matrix, and further inversely replicating the CDA pad by the intaglio Si platform prepared by photolithography and deep reactive ion etching. The acquired CDA-based ECG electrode has mushroom-like pillar arrays with an aspect ratio greater than three over a large area (Figure 1a). When used in ECG signal monitoring, the graphene-based CDA with ~100 $\Omega\cdot$cm low volume resistance enables clear ECG waveform display (P, QRS, and T waves), and its high, ~1.3 N/cm$^2$ normal adhesion force ensures it can achieve more than 30 repeated adhesions, even on rough of the human body. Most significantly, the CDA electrode is waterproof, dustproof, and reusable, unlike conventional ECG electrodes. Based on its high-aspect-ratio micropillars, the gecko-like surface of the electrode is super-hydrophobic (Figure 1b), which means dust on the electrode can simply be washed out without damaging its structure.

### 2.1.2. Amphibian- and Octopus-Inspired ECG Electrodes

Human skin is a rough surface with a certain degree of wetness, and ECG electrodes are prone to sweat interference during long-term wearing [37]. The adhesion systems of amphibians can enhance the adhesion forces on a wet or rough surface through their hexagonal structure [38]. Moreover, the convex suction cups of an octopus enable strong

adhesion to various surfaces in dry and wet conditions [39]. Inspired by the microchannel network in the toe pads of tree frogs and convex cups in the suckers of octopi, Kim et al. proposed a water-drainable and air-permeable graphene ECG electrode [40]. Based on silicon molds with micro-hole patterns, the frog- and octopus-inspired structures were replicated using PDMS. Later, the PDMS structure was covered by reduced graphene oxide (rGO) nanoplatelets through spray-coating (Figure 1c). Inspired by rain frogs that can attach to rough surfaces by routing water through their pads with hexagonal channels, electrodes with hierarchical architecture show high peeling energy on wet surfaces. To further increase wet adhesion of the electrode in different directions, inspired by the protuberant structures of the octopus's suckers, microscale cylindrical holes were stamped and molded on the contact surfaces of the electrode. Due to the suction effect of the microstructure, the electrode patches realized 6.6 N·cm$^{-2}$ adhesion strength on dry surfaces, 5.3 N·cm$^{-2}$ on wet surfaces, and 4.5 N·cm$^{-2}$ in underwater conditions. After coating with ultrathin rGO nanoplates to form the conductive layer, the realized ECG electrode could monitor ECG signals on sweaty skin during exercise or even under flowing water.

### 2.1.3. Avian Nest-Inspired ECG Electrode

For ECG electrodes that need to be worn for a long time, their mechanical properties should adapt to the multiple demands of the complex wearable environment. Although graphene has intrinsic flexibility, chemical stability, and high electrical performance, the conductive networks formed by graphene can easily be cracked without a proper supporting framework. The mud-based bird nest has marvelous firmness due to the filiform saliva with polysaccharides excreted by the bird [41]. Inspired by the solidification process of the avian nest, Qiu et al. reported a durable and non-disposable graphene skin–electrode for long-term ECG signal monitoring [42]. By mimicking the architecture and fabricating process of the avian nest, polymer nanofibers (consisting of $CuCl_2$, SEBS, and phenolic resin) prepared by electrospinning were captured by a substrate with CVD monolayer graphene film. After annealing, the nanofibers are attached tightly to the graphene film through π–π interactions between phenol side groups and metal cations, which benefits the growth of the crystalline graphene structure. Finally, the whole graphene–polymer nanofibers structure was embedded in SEBS solution to form the electrodes (Figure 1d). The prepared electrode has a structure similar to that of an avian nest, with graphene–polymer-nanofiber-based conductive networks and semi-embedded SEBS for mechanical strength enhancement. When used as an ECG electrode, it shows high conductivity (~150 Ω/square) and a high signal-to-noise ratio (30 dB). Besides, the semi-embedded electrode shows excellent washability and repeatability during signal monitoring.

### 2.1.4. Self-Healing ECG Electrode

If contact ECG electrodes can completely mimic human skin, which is stretchable, compliant, and self-healing, long-term ECG monitoring will comfortable and the electrodes will be more durable. To realize a self-healing ECG electrode with excellent stretchability and flexibility beyond that of human skin, Pan et al. realized a bionic electrode, based on conductive hydrogels, constructed of a nerve-like nanonetwork [43]. Through incorporating proanthocyanins/reduced graphene oxide (PC/rGO) with a nerve-like nanonetwork into a glycerol–plasticized polyvinyl alcohol–borax (PVA–borax) hydrogel system, a PC/rGO/PVA hydrogel with ultrahigh stretchability (>5000%), excellent compliance (1 mm), and fast self-healing ability (3 s, 95.73%) was realized (Figure 1e). The excellent self-healing ability of the hydrogel electrode is mainly due to the glycerol, which provides many free borax-chelating sites for self-repair. Like the regeneration process of subcutaneous neural networks, the catechol groups on the PC/rGO surface form many crosslinked bonds with the borax, which makes the hydrogel electrode heal quickly.

When attached to human skin, the electrode had a high signal-to-noise ratio, and the characteristic peaks of the ECG signals were clearly defined (Figure 1f), without a wandering baseline. Besides, the hydrogel electrode fits well with human skin, the testers

felt comfortable, and it did not affect their motion. Moreover, its self-healing ability ensures it can heal itself quickly when damaged, which greatly improves the service life of the electrode, making it suitable for long-term ECG monitoring.

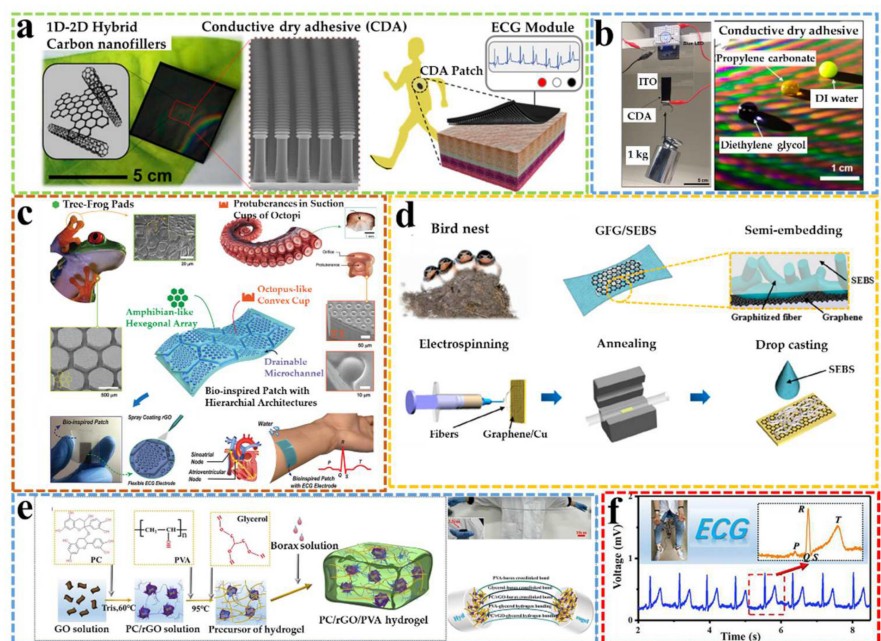

**Figure 1.** Bionic graphene ECG electrodes. (**a**) Gecko-inspired ECG electrodes. Reprinted with permission from Ref. [36]. Copyright 2016, American Chemical Society. (**b**) Photo of the strong adsorption of the CDA electrode (**left**) and liquid droplets on the gecko-inspired CDA electrode (**right**). Reprinted with permission from Ref. [36]. Copyright 2016, American Chemical Society. (**c**) Amphibian- and octopus-inspired hierarchical hexagonal microstructures for ECG monitoring. Reprinted with permission from Ref. [40]. Copyright 2016, American Chemical Society. (**d**) An avian-nest-inspired ECG electrode and its preparation process. Reprinted with permission from Ref. [42]. Copyright 2020, American Chemical Society. (**e**) Schematic of the preparation process (**left**) and the ultra-fast self-healing ability (**right**) of the PC/rGO/PVA hydrogel ECG electrode. Reprinted with permission from Ref. [43]. Copyright 2020, Elsevier. (**f**) The self-healing ECG electrode is used to detect ECG signals. Reprinted with permission from Ref. [43]. Copyright 2020, Elsevier.

## 2.2. Fabric-Based Graphene ECG Electrodes

A smart way to improve the wearable comfort of contact electrodes and make them suitable for long-term wear is to make functional daily-wear clothes or fabrics. Recently, with the rapid progress of fiber electronics, many fabric ECG electrodes have been developed [44]. Compared to conventional Ag/AgCl gel ECG electrodes, fabric electrodes have intrinsic advantages, including light weight, high gas permeability, and low cost, which make them more acceptable to the consumer market. However, most clothing fibers are electrically insulated and do not closely adhere to skin, which results in high contact impedance between electrode and skin, large motion artifact caused by relative movement between electrode and skin, and serious noise problems of collected ECG signal. Facing these challenges, much work has been carried out by combining graphene materials with fibers to endow the fabric ECG electrodes with high electrical conductivity without affecting its other advantages [45–53]. Meanwhile, some research on adhesive fabrics and electrode-wearing methods have been developed to improve contact between the fabric ECG electrode and skin. In this section, we will introduce several typical fabric ECG electrodes proposed in recent years, and discuss their characteristics.

### 2.2.1. Cotton-Textiles-Based Graphene ECG Electrode

At present, the most commonly used ECG electrodes are made of silver/silver chloride (Ag/AgCl) with a hydrogel layer. Although this kind of electrode is inexpensive and has small electrode–skin resistance, its gel layer may cause skin allergies. Further, during long-term ECG monitoring, the water content of the gel decreases, increasing electrode–skin contact resistivity. To overcome the skin allergy and gel-drying problems in long-term ECG monitoring, Xu et al. proposed a shuttle cotton textile-based ECG electrode [45]. Lightweight (130 g/m$^2$) commercial cotton textiles with fabric diameters ranging from 10–20 μm were used as the substrate for fabricating graphene-coated ECG electrodes. The surface of the textile was first coated with a polymer layer by heat transfer (180 °C, 10 s) to enhance the roughness of its surface and increase the adhesion between cotton fibers and graphene nanoplates. Then, screen-printing was used to tightly combine the conductive graphene patterns with the textile substrate, making the electrodes washable for long-term repeated use (Figure 2a). The prepared textile electrode shows a low resistance of ~104 Ω/square, and the acquired ECG signals are comparable with Ag/AgCl electrodes. Besides, the cotton textile substrate allows the electrode to continue to obtain high-quality ECG signals after washing. Moreover, as a high-throughput fabrication method, the produced cotton-textiles-based graphene ECG electrodes are low-cost, durable, and suitable for daily ECG monitoring for the consumer health market.

### 2.2.2. Silk-Based Graphene ECG Electrode

Electronic skins are a kind of newly developed skin contact electrode; they are lightweight, and soft like human skin, thereby allowing them to be attached to human skin without palpable sensation [46]. Silk fibroin is a natural protein material, produced by silkworms, with mechanical durability, tunable structures, and biocompatibility, making it an ideal material for making electronic skins. Wang et al. combined the silk with the thinnest electrical-conductive material, graphene, to develop an ultra-soft, tattoo-like ECG electrode [47]. Silkworm cocoons were used to prepare the silk fibroin (SF) solution. To give the electrodes self-healing function, Ca$^{2+}$ ions were added to the SF solution. The resulting SF/Ca$^{2+}$ mixture contained plenty of hydron and coordination bonds, which allows self-healing of the ECG electrode. By putting the graphene nanoplates into the SF/Ca$^{2+}$ mixture to prepare the graphene/silk fibroin/Ca$^{2+}$ (Gr/SF/Ca$^{2+}$) suspension, and using mask printing or direct writing to transfer the designed electrode pattern to the SF/Ca$^{2+}$ substrate, a tattoo-like, silk-based ECG electrode was realized (Figure 2c). The electrode can detect ECG signals with a high signal-to-noise ratio, and the continuous monitoring time can reach more than 10 h. With biocompatibility, self-healing ability, and good adhesion to human skin, the silk-based ECG electrode can be used for high-comfort, long-term, and high-quality ECG monitoring.

### 2.2.3. Nanofiber-Based Graphene ECG Electrode

Nanofibers are a kind of one-dimensional (1D) materials with unique physical and chemical characteristics [48]. They can form nano-scale networks with high porosity and ultra-light weight. Most significantly, their huge specific surface area can provide a large number of binding points that can combine with other functional materials to enable the nanofibers to have customized functions [49]. Electrospinning is a fast and simple way to produce nanofibers [50]. By applying an electrostatic field to the solution of the desired polymer, the corresponding polymer nanofiber can be "spun out". Based on electrospinning polyvinylidene difluoride (PVDF)/poly (3,4-ethylene dioxythiophene) polystyrene sulfonate (PEDOT: PSS) nanofibers, Huang et al. realized a carbon-nanofiber sensing electrode and applied it in smart clothing for ECG monitoring [51] (Figure 2c). A conductive film consisting of reduced graphene oxide and carbon black was used to collect the electrospun nanofibers. After electrospinning, the conductive film was combined with the nanofibers to create a nanofiber-based electrode with high conductivity (25 Ω/square) and high stability (can sustain over 3000 repeated uses). The nanofiber-based graphene ECG electrode

increases the contact area between the electrode and skin, and shows good hydrophobicity while realizing high stability and excellent electrical conductivity, making it suitable for long-term ECG monitoring. Moreover, its performance in intelligent clothing shows its excellent stability for monitoring ECG signals while running.

### 2.2.4. Nylon-Textile-Based Graphene ECG Electrode

It is time-saving and cost-effective to change insulation fabric materials into conductive materials using traditional cloth dyeing methods. Besides, the fabric ECG electrode realized by this method can be directly integrated into undergarments for daily ECG signal monitoring. Nylon has long been used as a reliable clothing material with low surface roughness, high strength, and excellent elasticity [52]. When impregnated and dried with a conductive dye, the conductive nylon composite can be used as a wearable fabric electrode. Yapici et al. proposed graphene-clad-nylon textile electrodes [53]. First, by using GO suspension as the dye and dipping the nylon textiles in it, nylon textiles fully impregnated with GO slurry were obtained. Then, the GO-impregnated nylon textile was thermally treated to form a uniform GO–nylon core-sheath structure. Finally, the GO–nylon fibers were reduced by hydrogen iodide (HI) to create a graphene–nylon conductive network (Figure 2d). To apply graphene–nylon electrodes to clothing and create healthcare garments, the graphene–nylon electrodes were cut into pieces and attached to clothing, including wristbands, neckbands, etc., to create daily, multi-lead ECG monitoring. In combination with wearable processing systems, real-time ECG signals were acquired by the nylon-textile-based graphene ECG electrodes and transferred to a backend system wirelessly for further analysis. This work created a complete prototype of the wearable garment based on fabric-graphene ECG electrodes.

### 2.3. Biodegradable Graphene ECG Electrodes

As people's demand for daily wearable health monitoring increases [54], consumption of health monitoring products such as ECG electrodes will also increase greatly. However, due to cost, durability, and other factors, wearable devices such as the widely used commercial gel electrodes are mostly made of non-degradable materials. To meet the environmental protection needs, this section will introduce examples of high-performance, biodegradable graphene ECG electrodes.

### 2.3.1. Ultra-High-Skin-Conforming and Biodegradable Graphene ECG Electrode

Polyvinyl alcohol (PVA) is a water-soluble and biodegradable organic material very suitable for flexible and environmentally friendly wearable electronics [55]. Since it does not conduct electricity, it is usually used as a substrate for conducting materials. Wu et al. proposed a PVA/graphene flexible ECG electrode fabricated via double transfer [56] (Figure 2e). Compared with previous techniques that require heating and etching, the new process is efficient and low-cost. The resulting graphene electrode has high flexibility (Young's modulus of 8.598 MPa) and can detect ECG signals with a high signal-to-noise ratio (60 dB). The ECG electrode can simply be delaminated by water in 150 s, which offers a good solution for biodegradable wearable electronics.

### 2.3.2. Graphene–PHA ECG Electrode

Polyhydroxyalkanoate (PHA) is a kind of aliphatic polyester bioplastic that is accumulated by Ralstonia eutropha [57]. With PHA nanofibers and graphene, Suvarnaphaet et al. created a graphene–PHA ECG electrode that is biocompatible and can be completely biodegraded by microbes in terrestrial environments [58]. The PHA nanofibers were produced by electrospinning and served as the scaffold for graphene. The conductive graphene was derived by hydrazine-hydrate-reduced graphene oxide, and was applied to the PHA scaffold to form the graphene–PHA composite by vacuum filtration. After being compressed by a roller to boost its conductivity, the graphene–PHA ECG electrode showed a high-performance ECG monitoring capability. Besides, although the electrode is biodegradable, it can be kept for 1~2 years at the right humidity and oxygen levels.

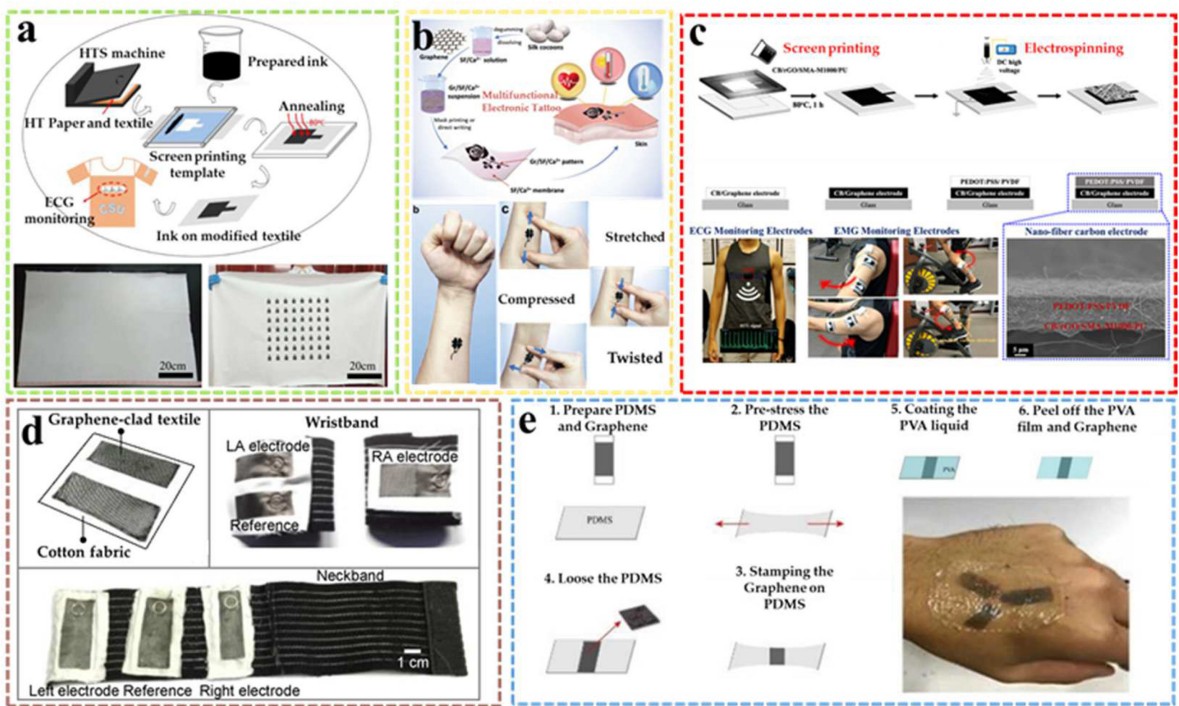

**Figure 2.** (**a**) The fabrication process (**left**) and photos (**right**) of the cotton-textiles-based graphene ECG electrode. Reprinted with permission from Ref. [45]. Copyright 2020, IOPscience. (**b**) The fabrication process of the Gr/SF/Ca$^{2+}$ tattoo electrode and its stable adhesion on human skin. Reprinted with permission from Ref. [47]. Copyright 2020, Wiley. (**c**) The preparation process of the nanofiber-based graphene ECG electrodes (**above**), its applications in ECG and EMG monitoring (**lower left**), and its SEM images (**lower right**). Reprinted with permission from Ref. [51]. Copyright 2021, American Chemical Society. (**d**) The graphene–nylon-based textile electrodes and their integration in a garment. Reprinted with permission from Ref. [53]. Copyright 2017, MDPI. (**e**) Preparation processes of the PVA/graphene biodegradable electrode. Reprinted with permission from Ref. [56]. Copyright 2020, International Journal of Engineering Research and Applications.

### 2.4. Laser-Induced/Scribed Graphene ECG Electrodes

Producing graphene rapidly and in large quantities is a challenging problem that must be solved if graphene ECG electrodes are to be implemented on a large scale. Laser induction and laser scribing are rapid and low-cost graphene production methods that have developed rapidly in recent years [59,60]. Not only can laser-induced graphene (LIG) and laser-scribed graphene (LSG) be prepared rapidly and at low cost on a large scale, but, most importantly, the preparation of graphene and fabrication of corresponding graphene electronics can be accomplished simultaneously. Based on these techniques, a series of laser-induced/scribed graphene ECG electrodes have been proposed [61–67], whose patterns can be designed flexibly and processed quickly according to attachment position, impedance, and other requirements, which has good application potential.

#### 2.4.1. Porous LIG ECG Electrode

When preparing LIG electrodes, a thin film polyimide (PI) was used as a substrate material to obtain different graphene structures. When the laser intensity applied to PI is high enough, the laser ablation carbonizes the PI, and the resulting nitrogen and carbon oxide gases spill out to form a porous structure on the surface of LIG [61]. The resulting porous LIG has high electrical conductivity and flexibility, and the porous surface helps increase the contact area between the skin and electrode, suitable for the electrode conductive layer. Based on porous LIG, Toral et al. prepared a LIG ECG electrode using a 2.4 W CO$_2$ laser [62]. The graphene properties of laser-ablated carbides were proven by

Raman spectroscopy. The sheet resistance of the porous LIG electrode is 360 $\Omega/cm^2$, and the contact impedance of the electrode is 700 $\Omega$, similar to commercial gel electrodes. When used to detect ECG signals, typical ECG features can be well recognized.

### 2.4.2. Water-Stable LSG ECG Electrode

To produce graphene devices on a large scale, it is very convenient to process graphene materials in liquid. However, as a hydrophobic material, graphene easily agglomerates in solution, and it is not easy to disperse it evenly, which increases the difficulty of the graphene liquid phase process [63]. Besides, the high cost of high-quality graphene remains an important limitation for its application. Therefore, GO plays an important role as a low-cost and water-dispersible graphene material. It can be processed as a suspension and then reduced to conductive rGO by several means after the device's previous processing. Traditional thermal or chemical reduction methods may release toxic and harmful substances, require large energy consumption, and take a long time. By using a laser to reduce GO, the conductive electrode can be prepared quickly and energy-efficiently with customizable patterns. Based on this method, Murastov et al. proposed a water-stable and flexible LIG electrode for ECG monitoring [64]. They used polyethylene terephthalate (PET) as the substrate for the GO suspension's drop-casting. After drying on a hot plate, a thin-film GO layer was prepared on the surface of PET. Then, a 405 nm laser with a maximum power of 1 W was used to induce the rGO conductive pattern (Figure 3a). The resulting LIG electrode shows excellent stability in solutions with different pH and realizes a long-term wearing over 100 h. This work demonstrates that the LIG electrode can maintain certain stability when exposed to human sweat, ensuring long-term monitoring of ECG signals.

### 2.4.3. Polyaziridine-Encapsulated LIG ECG Electrode

Although laser induction is an effective way to produce graphene electrodes, the realized LIG flakes are not well-interconnected due to the gap between the laser-scribed lines [61]. This results in uneven distribution of conductivity across the electrode, reducing overall conductivity. When used for daily ECG monitoring, cracks on the electrode surface will be further aggravated by movement, and the quality of the detected ECG signal will be reduced. To solve this problem, Zahed et al. proposed a polyaziridine-encapsulated, 3D porous LIG electrode [65]. In this work, polyaziridine was used as a linker inside the porous LIG to interconnect the graphene flakes. As shown in Figure 3b, the amine groups in polyaziridine can covalently bind with many oxygen-containing groups of LIG, which ensures the LIG flakes are electrically connected under tension or bending. The ECG signals collected by the electrodes are of good quality both in static and dynamic conditions. Besides, due to the large specific surface area of 3D porous LIG (340 $m^2/g$), the ECG electrode can achieve a signal-to-noise ratio comparable to conventional Ag/AgCl gel electrodes (13.5 dB) with only half of the area.

### 2.4.4. Stretchable LIG ECG Electrode

Usually, LIG is fabricated and instrumented simultaneously on PI substrate. This method is very simple and efficient, and the resulting LIG electrode has good electrical conductivity. However, due to the limited flexibility of the PI substrate [66], the LIG electrode can only bend with the PI substrate, not stretch freely. This greatly limits the use of LIG electrodes because, as the surface of the human body is soft, uneven, and movable, it is difficult for the PI substrate to merge and move tightly with the skin. This situation further affects comfort and the quality of collected ECG signals. To solve the disadvantages of the PI substrate while retaining the excellent ECG monitoring performance of the LIG. Zhang et al. proposed a stretchable LIG ECG electrode by replicating the LIG patterns with ultrathin and highly flexible polystyrene-block–poly(ethylene-ran-butylene)-block–polystyrene (SEBS), packaged into a highly stretchable layer-by-layer structure [67]. As shown in Figure 3c, the LIG was first produced by a 10.6 $\mu m$ $CO_2$ laser on a PI film. Then, 3,4-ethylene dioxythiophene (EDOT) was assembled on MXene-$Ti_3C_2$ to give the LIG

electrode temperature- and strain-sensing ability, improve its conductivity, and enhance its viscosity with SEBS to facilitate LIG transfer and packaging. Furthermore, an MXene-$Ti_3C_2$@EDOT layer was transferred onto the LIG by electrophoretic deposition. Finally, the MXene-LIG layer was replicated from the PI substrate to the SEBS and encapsulated. The resulted high flexibility and high conductivity electrode cannot only detect ECG signals but can also monitor temperature and motion. When monitoring ECG signals, the MXene-LIG electrode realizes significantly lower contact impedance (51.08 kΩ at 10 Hz) than the commercial Ag/AgCl gel electrode, enabling high-quality detection of ECG signals. Moreover, the stable and highly flexible electrode realizes a higher signal-to-noise ratio during long-term and real-time ECG monitoring, which provides a promising development direction for graphene ECG electrodes.

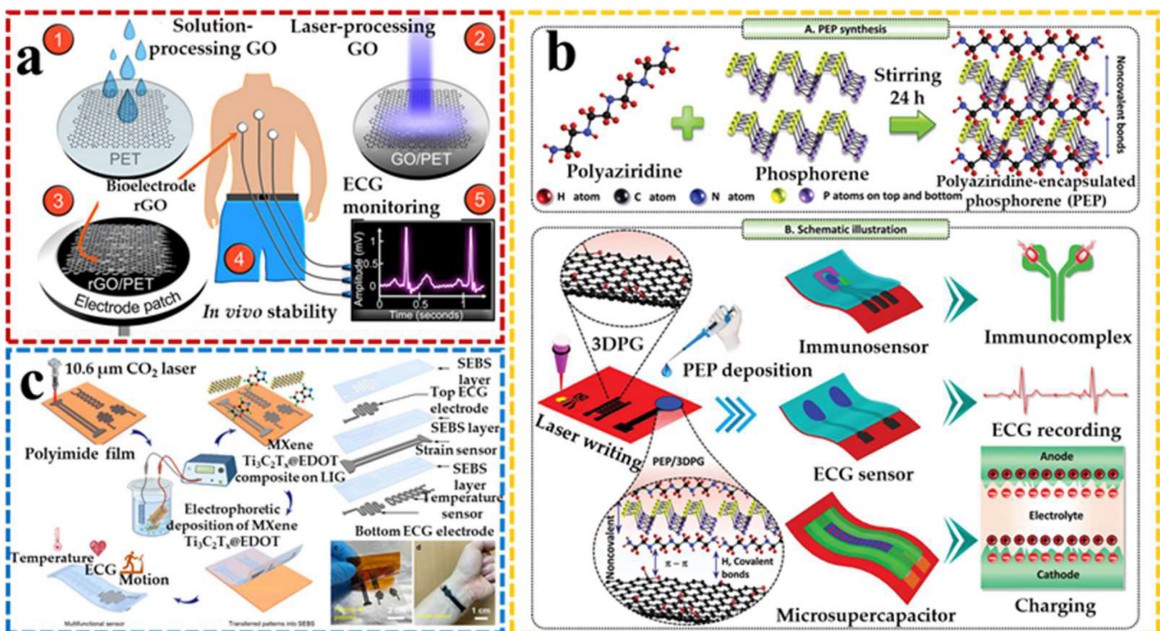

**Figure 3.** Laser-induced/scribed graphene ECG electrodes. (**a**) Schematic illustration of the preparation process of the LIG ECG electrode. Reprinted with permission from Ref. [64]. Copyright 2020, Elsevier. (**b**) Preparation of the polyaziridine-encapsulated phosphorene composite (**above**) and the schematic diagram of the ECG electrode, immunosensor, etc. based on 3D porous LIG (**below**). Reprinted with permission from Ref. [65]. Copyright 2021, Wiley. (**c**) Schematics of the fabrication process and photos of the LIG–SEBS-based multifunctional sensors. Reprinted with permission from Ref. [67]. Copyright 2022, Springer Nature.

### 2.5. Modified-Graphene-Based ECG Electrodes

Appropriate modifications can give materials superior properties, and graphene is no exception as a material for ECG electrodes. In this section, we will introduce some typical graphene modification methods, including physical doping and chemical modification, that give graphene electrodes better performance.

### 2.5.1. Chemically Modified Graphene-CNT ECG Electrode

Chemical modification and doping are good methods to improve the electrical properties of graphene electrodes, and if the two can be combined, better results may be achieved. Based on chemically modified graphene and functionalized multiwalled carbon nanotubes (MWCNT), Hossain et al. developed a paper-based flexible ECG electrode [68]. Compared to CVD growth of graphene, the use of chemically modified GO (CG) is a cheaper and more efficient way to obtain graphene materials. To improve electrical conductivity between CG flakes, a conductive network is formed by MWCNT. Since CG has a lot of oxygen groups on its surface, by using carboxylic groups to functionalize the MWCNT, the modified MWCNT

binds more tightly to CG. CG acts as the carrier of the MWCNT, and the CNTs anchor the CG nanoflakes. The modified surface groups make them difficult to aggregate, which forms a more uniform conductive work (Figure 4a). By transferring the CG-MWCNT composite to the paper substrate, the flexible ECG electrode shows low sheet resistance (75 $\Omega$/square) and low contact impedance (45.12 k$\Omega$ at 100 Hz).

### 2.5.2. Environmentally Friendly Chemically-Modified Graphene ECG Electrode

Chemical reduction of GO using biocompatible materials is an environmentally friendly way to produce graphene electrodes. Hossain et al. used glucose as the reducing agent to fabricate biocompatible, glucose-treated, chemically modified graphene (GCG) through a solvothermal technique (180 °C, 2 h in a convection autoclave) [69]. After acidic treatment (1 M acetic acid, 5 h), the GCG was transferred to a contamination-free CG suspension. Then, through filtration, the GC film was cast onto a nylon filter paper to create an ECG electrode (Figure 4b). The achieved graphene electrode can detect the characteristic peaks of the ECG signal. Besides, it provides an environmentally friendly method for producing graphene ECG electrodes.

### 2.5.3. Intercalation Doping Graphene ECG Electrode

If monolayer graphene is directly applied to ECG electrode, it very easily cracks due to external stress [70], resulting in rapid increases in electrode resistivity. If multiple layers of graphene are used to relieve stress by sliding between graphene sheets, the electrical connection between graphene sheets is not guaranteed. To take full advantage of the sliding between graphene sheets while ensuring good interlayer interconnection, Du et al. used molybdenum chloride ($MoCl_5$)-intercalated bilayer graphene (Mo-BLG) to prepare a graphene ECG electrode [71]. To create a larger-scale, high-quality, high-conductivity, and stretchable graphene ECG electrode, they used CVD graphene and transferred it by a wet method to form the layer-by-layer graphene structure. Then the few-layer graphene was put into an $N_2$-filled tube with $MoCl_5$ and $MoO_3$ powder and heated. The resulting Mo-BLG was transferred to the SEBS substrate and encapsulated (Figure 4c). Due to the self-barrier doping effect, the Mo-BLG can maintain stable impedance for a long time, and the interlayer doping can lubricate and interconnect the graphene sheet for good stretching properties. With a low sheet resistance (40 $\Omega$/square) and higher signal-to-noise ratio than commercial Ag/AgCl electrodes, the electrode, which can withstand stress while wearing, is suitable for long-term and reliable ECG monitoring.

### 2.5.4. Glycerol-Modified Graphene-Oxide ECG Electrode

Compared with graphene, the preparation of GO is simpler and cheaper. It is an effective way to reduce the cost of graphene electrodes by transforming GO that can be prepared in large quantities into graphene materials. Based on esterification of glycerol (Gl) and a polyvinyl alcohol (PVA) polymer host, Maher et al. proposed a GO/Gl/PVA ECG electrode [72]. First, GO was prepared by a modified Hummers' method and was mixed with Gl at pH 1–3. After a full reaction, the conductive GO/Gl was further mixed with PVA to form an elastic conductor. Finally, the GO/Gl/PVA was put into the ECG electrode mold (Figure 4d). As the carrier of the GO/Gl, PVA enables conductive reduced GO sheets to be evenly distributed. Besides, since PVA is a water-soluble polymer, after applying the electrodes to skin using water, the resulting ECG electrode shows high flexibility, good adhesion, and low impedance between the electrode and human skin. The detected ECG signal has a high signal-to-noise ratio.

### 2.5.5. Hybrid Copper Nanoparticles–Graphene Oxide ECG Electrode

In ECG electrodes, graphene cannot only be used as a conductive material, but also as auxiliary material to help conductive materials achieve better performance. Copper nanoparticles (CuNPs) are inexpensive metal particles with high conductivity and are harmless to skin [73]. However, their large surface area makes them vulnerable to oxi-

dation when exposed to air, resulting in a rapid reduction of their electrical conductivity. Besides, agglomeration leads to uneven conductivity in CuNP-based electrodes. To improve their air stability and dispersion uniformity, Huang et al. used CuNPs, graphene oxide, and polyethylene glycol-trimethylolpropane-trimellitic anhydride (PTT) to fabricate a CuNP/PTT/GO composite ECG electrode [74]. As the stabilizer of CuNPs, PTT absorbed the CuNPs through noncovalent van der Waals force and ionic-charge interactions of the noncovalent bond. Most significantly, the epoxy groups on GO can form an electrostatic attraction force to the $Cu^{2+}$, which makes the CuNPs realize nanoscale stabilization (Figure 4e). The organic/inorganic hybrid GO/CuNPs shows long-term stability and ultrahigh conductivity (0.906 $\Omega$/square after 30 days of storage), which are suitable for long-term wearable ECG monitoring.

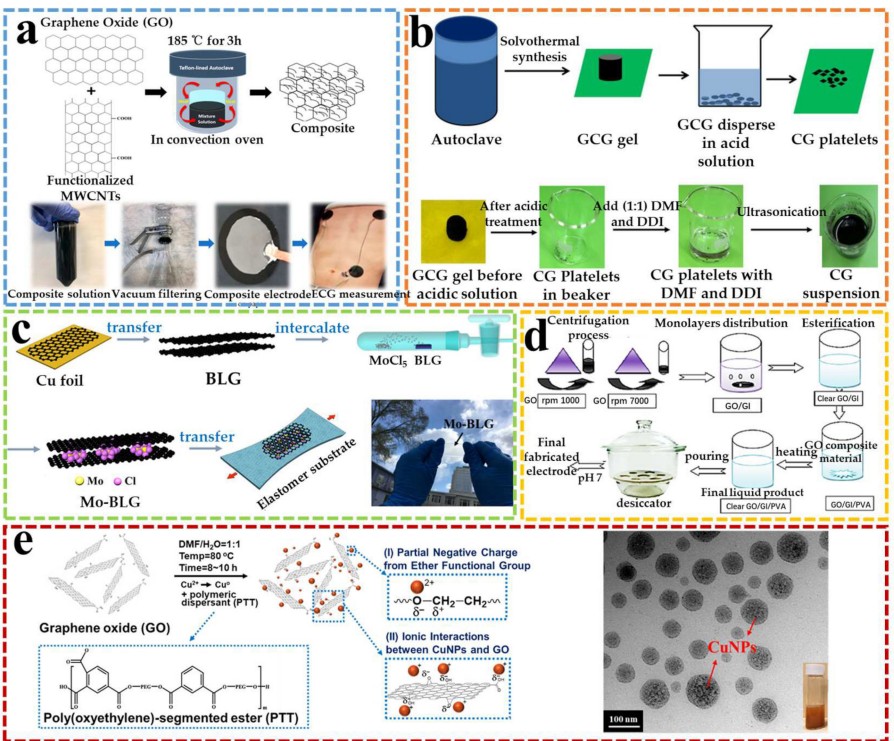

**Figure 4.** Modified-graphene based electrodes. (**a**) The schematic illustration of the preparation processes (above) and photos (below) of chemically-modified graphene–CNT ECG electrodes. Reprinted with permission from Ref. [68]. Copyright 2019, MDPI. (**b**) Preparation processes of the environmentally friendly, chemically modified graphene ECG electrode. Reprinted with permission from Ref. [69]. Copyright 2017, IOPscience. (**c**) Fabrication processes and photo of the intercalation doping graphene ECG electrode. Reprinted with permission from Ref. [71]. Copyright 2020, American Chemical Society. (**d**) Preparation of the glycerol-modified, graphene-oxide ECG electrode. Reprinted with permission from Ref. [72]. Copyright 2022, IOPscience. (**e**) Fabrication processes of hybrid copper nanoparticle–graphene oxide and its photos. Reprinted with permission from Ref. [74]. Copyright 2022, MDPI.

## 2.6. Sponge-Like Graphene ECG Electrodes

Porous, soft, and fluffy spongy graphene materials have unique advantages as skin-contact electrodes because of their large surface area, light weight, and flexibility. In this section, we will introduce some typical graphene ECG electrodes with sponge-like structures and show their advantages in ECG monitoring.

### 2.6.1. Graphene Sponge ECG Electrode

A graphene sponge is a highly compliant and porous material that is structurally stable and flexible. Asadi et al. proposed a graphene sponge ECG electrode fabricated by freeze-casting [75]. First, GO prepared by Hummers' method was partially reduced by ascorbic

acid. Second, the mixture was boiled, followed by anisotropic freeze casting and a second reduction process. Finally, the acquired hydrogel was freeze-dried and annealed to create the sponge-like structure of graphene. The graphene sponge is ultra-lightweight (5 mg/cm$^3$), has a stable cellular structure and high resilience (with stable elastic properties over 1000 cycles of compression), and is ultra-soft (its elastic modulus is 1,000,000 times lower than that of PDMS). As an ECG electrode, the graphene elastomer electrode shows a significantly low skin–electrode impedance and a high signal-to-noise ratio.

### 2.6.2. PVA/PED/GO Sponge ECG Electrode

Gel ECG electrodes lose water over time, resulting in increased impedance and poor signal quality [76], which is not suitable for long-term ECG monitoring. However, hydrogel is still an efficient basis for electrodes, as it gives electrodes flexibility and helps the electrode materials mix evenly. To make full use of the advantages of hydrogel, give it a structure more suitable for electrode application, and solve the problem of easy water loss, Xiao et al. proposed a three-dimensional network structured polyvinyl alcohol/polyethylene glycol/graphene oxide (PVA/PED/GO) gel through cyclic freezing–thawing [77]. During the freezing process, a complete polymeric net was formed by dense hydrogen bonds, giving the composite good electrical conductivity and high hydrophilicity and elasticity (Figure 5a). Unlike the commercial gel electrode, the PVA/PED/GO electrode did not show signs of allergic reaction during in vitro testing.

### *2.7. Invasive Graphene ECG Electrodes*

While a wide variety of epidermal graphene ECG electrodes offer a low-cost, high-comfort, and non-invasive solution to long-term ECG monitoring, there are still limitations in the quality of the collected ECG signals. For example, epidermal electrodes tend to move relative to the skin in daily wear, which may introduce motion artifacts [78]. At the same time, non-invasive electrodes are more susceptible to external noise interference, which reduces the signal-to-noise ratio [79]. In addition, skin conditions, such as sweat, skin thickness, and cleanliness will greatly affect electrode–skin impedance [80]. To address the disadvantages of epidermal electrodes, some invasive graphene electrodes have been developed to produce higher-quality ECG signals.

### 2.7.1. Microprobe Graphene ECG Electrode

For higher-quality ECG signal monitoring, ECG electrodes cannot only be deployed on the body's surface, but can be used for invasive monitoring or even monitoring ECG signals from various bio-objects. Chen et al. proposed a flexible and hydrophilic graphene ECG microprobe to realize high-resolution ECG signal monitoring for various creatures [81]. Through microelectromechanical (MEMS) technology, graphene grown by chemical vapor deposition (CVD) was transferred to a SU-8 substrate. After being encapsulated by PDMS and treated with steam plasma, the graphene microprobe showed flexibility, biocompatibility, and hydrophily, which allow it to form a low-impedance electrical contact with the electrolytes of living organisms (Figure 5b). When used in ECG monitoring of zebrafish and crayfish, the electrode shows a low noise level (4.2 $\mu V_{rms}$) and high signal-to-noise ratio (27.8 $\pm$ 4.0 dB), which shows potential in long-term, in vivo ECG monitoring.

### 2.7.2. Radiolucent Graphene ECG Electrode

Traditional implantable ECG electrodes are not electromagnetically transparent, which causes implantable ECG electrodes to interfere with X-ray imaging [82]. As an electromagnetically transparent material, graphene's good electrical properties, biocompatibility, and flexibility also make it suitable for use as an implantable electrode [83]. Bong, et al. proposed a radiolucent implantable graphene ECG electrode [84]. The electrode was fabricated on a biocompatible, CVD-prepared parylene C substrate. Monolayer graphene was CVD prepared and transferred onto the substrate, forming 16 graphene electrode sites by reactive ion etching. After wiring by Ti/Au/Pt, the electrode was encapsulated by another parylene C

layer (Figure 5c). In vivo tests were carried out on a dog since humans and dogs have similar ECG characteristics. The test result show the radiolucent graphene ECG electrode has good biocompatibility and ECG monitoring performance. Besides, it does not interfere with the quality of X-ray imaging performed during ECG monitoring. The radiolucent graphene ECG electrode shows good application prospects in the field of implantable ECG monitoring.

### 2.8. Other Graphene ECG Electrodes

In addition to the typical graphene ECG electrodes mentioned above, there are some other graphene electrodes with unique preparation methods or characteristics that will be introduced in this section.

### 2.8.1. GNR ECG Electrode

Graphene nanoribbon (GNR) is an important type of graphene material. Its various edge structures can bring it rich optical, electrical, magnetic, and semiconductor characteristics [85]. Prasad et al. proposed a GNR electrode for ECG signal acquisition [86]. They synthesized GNR with good electrical and mechanical properties by unzipping the MWCNT through oxidation and reduction. Then, the biocompatible and water-soluble PVA base material was mixed with GNR to form a PVA/GNR composite, which was finally implemented in an ECG electrode. The PVA/GNR electrode shows excellent performance (59.99 dB signal-noise-ratio, 348.62 mV R-peak value) with only 1 wt% GNR, which shows the potential of GNR applied to low-cost graphene ECG electrodes.

### 2.8.2. Monolayer Graphene ECG Electrode

The most direct and simple method to fabricate graphene ECG electrodes is by using a piece of monolayer graphene. Besides, using monolayer graphene can give full play to the ultra-thin characteristics of graphene to make ultra-thin ECG electrodes. Celik et al. developed an earphone-type graphene ECG electrode [87]. The CVD graphene was transferred to an Ag substrate through a sacrificial layer. The thickness of the graphene layer was around 0.37 nm. When used for ECG monitoring, motion noise is reduced because the electrode is located in the ear.

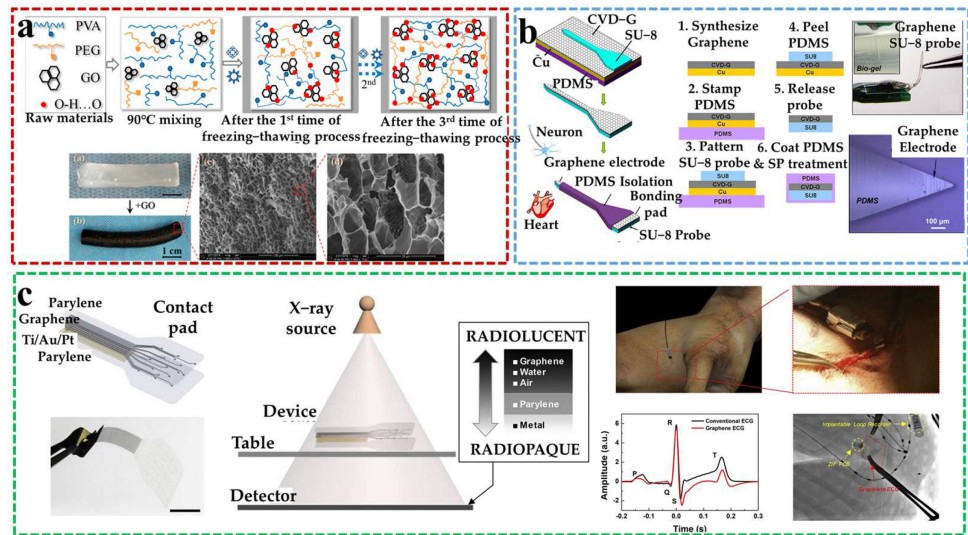

**Figure 5.** (**a**) Schematic illustrations of the preparation of the PVA/PED/GO sponge ECG electrode (**above**) and its digital images (**below**). Reprinted with permission from Ref. [77]. Copyright 2017, MDPI. (**b**) Fabrication processes and photos of the microprobe graphene ECG electrode. Reprinted with permission from Ref. [81]. Copyright 2013, Elsevier. (**c**) Fabrication processes and photos of the radiolucent graphene ECG electrode and its application in an animal. Reprinted with permission from Ref. [84]. Copyright 2019, Elsevier.

## 3. Discussions and Outlook

In the above sections, we introduced typical graphene ECG electrodes developed in recent years. The graphene ECG electrodes were classified according to their design concept, structural characteristics, preparation methods, and material properties. Facing the need for daily long-term cardiovascular health monitoring, graphene ECG electrodes take advantage of graphene's flexibility to improve comfort and conformability to human skin, enhance electrical conductivity to ensure the quality of collected ECG signals, and improve their stability to enable long-term use. In Table 1, we summarize the main electrical and mechanical characteristics including electrode–skin contact impedance/electrode resistance and the flexibility of the graphene electrodes introduced above, label the electrodes as dry or wet, and show their decided advantages.

**Table 1.** Electrical and mechanical properties of the flexible, graphene-based ECG electrodes mentioned above, with their decided advantages.

| Categories | Electrodes (Dry/Wet) | Sheet Resistance or Contact Impedance | Flexibility | Advantages |
|---|---|---|---|---|
| Commercial electrode | Ag/AgCl gel (wet) [88] | 15 kΩ at 1 kHz | Depends on its substrate | Widely used, low cost |
| Bionic graphene ECG electrodes | Gecko-inspired electrode (dry) [36] | ~100 Ω·cm | Stretchable and bendable | ~1.3 N/cm$^2$ adhesion force |
| | Amphibian- and octopus-inspired electrode (dry) [40] | / | Can be attached to skin conformally | 6.6 N/cm$^2$ adhesion force, waterproof |
| | Avian-nest-inspired electrode (dry) [42] | ~150 Ω/square | More than 10 attaching and detaching cycles | 30 dB signal-to-noise ratio |
| | Self-healing electrode (wet) [43] | / | >5000% stretchability | High healing efficiency (3 s, 95.73%) |
| Fabric-based graphene ECG electrodes | Cotton-textiles-based electrode (dry) [45] | ~104 Ω/square | Over 2000 bending cycles | 130 g/m$^2$ lightweight |
| | | | | Low preparation complexity and cost |
| | Silk-based electrode (wet) [47] | 96 ± 8 Ω/square | Can be stretched, compressed, and twisted | High healing efficiency (3 s, 100%) |
| | Nanofiber-based electrode (dry) [51] | 25 Ω/square | More than 3000 repeated uses | Hydrophobic |
| | Nylon-textile-based electrode (dry) [53] | 87.5 kΩ to 55 kΩ in the 10–50 Hz range | Flexible | Have been integrated into garments |
| Biodegradable graphene ECG electrodes | Ultra-high skin-conforming electrode (dry) [56] | 75 Ω/square, 45.12 kΩ at 100 Hz | Young's modulus of 8.598 MPa | 60 dB signal-to-noise ratio |
| | Graphene–PHA (dry) [58] | ~20 kΩ along the 25 mm × 5 mm stripe | Flexible | Can be kept for 1~2 years |
| Laser-induced/scribed graphene ECG electrodes | Porous LIG electrode (dry) [62] | 360 Ω/cm$^2$, 700 Ω | Flexible | Cost-effective |
| | Water-stable LSG (dry) [64] | ~4 kΩ | Flexible | 70 dB signal-to-noise ratio, 100 h monitoring in harsh environment |
| | Polyaziridine-encapsulated electrode (dry) [65] | 98.2 kΩ at 4 Hz to 90.35 kΩ at 1 kHz | Bendable | 13.5 dB signal-to-noise ratio |
| | Stretchable LIG electrode (dry) [67] | 51.08 kΩ at 10 Hz | High strain sensitivity of 2075 | 20.14 dB initial signal-to-noise ratio |
| Modified-graphene-based ECG electrodes | Chemically-modified graphene-CNT electrode (dry) [68] | 75 Ω/square, 45.12 kΩ at 100 Hz | Flexible | High comfortability and high sensitivity |
| | Environmentally friendly chemically-modified (dry) [69] | / | Flexible | ECG signal was received after 30 days of monitoring |
| | Intercalation doping electrode (dry) [71] | 40 Ω/square | Can sustain 80% strain | Durable over 2000 cycles at 30% strain |

**Table 1.** *Cont.*

| Categories | Electrodes (Dry/Wet) | Sheet Resistance or Contact Impedance | Flexibility | Advantages |
|---|---|---|---|---|
| | Glycerol-modified graphene-oxide electrode (dry) [72] | 30.22 µs skin conductivity | Plastically | Long-term stability |
| | Hybrid copper nanoparticle–graphene oxide electrode (dry) [74] | $8.12 \times 10^{-2}$ Ω/sq | Bendable | Air-stable, washable |
| Sponge-like graphene ECG electrodes | Graphene sponge electrode (dry) [75] | 230 kΩ at 20 Hz–200 kΩ at 40 Hz | 1,000,000 times lower elastic modulus than PDMS | 5 mg/cm$^3$ ultra-lightweight |
| | PVA/PED/GO sponge electrode (wet) [77] | 3.545 kΩ | Nearly 1000% breaking strain | Highly hydrophilic |
| Invasive graphene ECG electrodes | Microprobe graphene electrode (dry) [81] | / | Bendable | Low noise level (4.2 µV$_{rms}$) and high signal-to-noise ratio (27.8 ± 4.0 dB) |
| | Radiolucent graphene electrode (dry) [84] | 76 Ω/square | Bendable | Does not interfere with X-ray imaging |
| Other graphene ECG electrodes | GNR electrode (dry) [86] | Constant impedance of 250 kΩ over the frequency range 10 Hz–1 kHz | Flexible | 59.99 dB signal-noise-ratio, water-soluble |
| | Monolayer graphene electrode (dry) [87] | 5.5 kΩ at 1 kHz | / | 0.37 nm thickness |

Through the joint efforts of many researchers, graphene ECG electrodes have developed rapidly in recent years. Their electrical properties, mechanical properties, and stability have been greatly improved. Of course, graphene ECG electrodes still have a lot of room for improvement, and many scientific and engineering issues need to be resolved before they can be used on a scale similar to today's commercial Ag/AgCl gel electrodes. Next, based on the recent research work towards more comfortable and high-performance graphene ECG electrodes and related systems, our views on the possible development direction of graphene ECG electrodes are shared in the following paragraphs.

### 3.1. Improvement of Electrical Conductivity and Durability

When flexible graphene electrodes are used for ECG monitoring on the skin, although the electrode itself is flexible, more and more cracks appear in the graphene after repeated stretching, increasing electrode impedance. This will further affect the monitoring quality of ECG signals under long-term use. To solve this problem, a flexible ECG electrode based on silver nanowire bridging the graphene has been proposed [89]. By imitating the plasmodesmata in higher plants, silver nanowires (AgNWs) were used to bridge laser-scribed graphene oxide (LSG) nanoflakes (Figure 6a). In this way, LSG nanoflakes remain interconnected by AgNWs after the flexible electrode is stretched during wearing, which stabilizes impedance and improves durability, increasing suitability for long-term ECG monitoring. In addition to the above doping method, electrical conductivity and durability can also be improved by improving the physical structure of the electrode. As shown in Figure 6b, styrene–butadiene foam was used as the skeleton structure of the graphene electrode. The realized three-dimensional porous graphene electrode is made of soft and durable styrene–butadiene rubber micro-balls wrapped with graphene nanoflakes, creating a dense and resilient conductive network. In addition, the various modified graphene electrodes mentioned above are also typical methods to improve electrical conductivity and durability [68,69,71,72,74].

### 3.2. Large-Scale and Low-Cost Fabrication of Graphene Electrodes

The widely used commercial Ag/AgCl gel electrode has the characteristics of large-scale and low-cost manufacturing. Graphene ECG electrodes have achieved more excellent performance in monitoring ECG signals; the achievement of large-scale, high-quality preparation will profoundly impact the extent of their application. For this challenge, we tried to achieve large-scale preparation of graphene electrodes by a roll-to-roll method

(R2R) [15]. Compared to the traditional CVD method, the open-R2R CVD method does not need surface cleaning, polishing, or heat treatment of the copper substrate. The large area of graphene film (several meters long and 15 cm wide) was laminated onto an ethylene vinylacetate/polyethylene terephthalate EVA/PET substrate by rollers (Figure 6c). The resulting graphene/EVA/PET electrode has a lower electrode–skin impedance (less than 1 kHz) than commercial gel electrodes with high durability (can sustain 10 wash cycles). This provides an idea for how to prepare high-quality graphene ECG electrodes at a large scale and low cost.

In addition, to achieve low-cost and high-performance ECG electrodes, we also tried to produce ECG electrodes using waste materials. The COVID-19 pandemic has led to the mass discarding of medical masks, which are made of non-woven fabrics that do not degrade easily and may not be suitable for reuse due to contamination [90]. However, after disinfection, the non-woven fabric can be an excellent substrate for graphene electrodes since it has good flexibility, durability, comfort, and air-permeability. As shown in Figure 6d, after being dip-coated by graphene nanoflakes, a breathable, non-woven electrode can be made into the desired pattern. Holes are punched in it to ensure the sticky, non-woven fabric at the back increases adhesion between the electrode and skin, which decreases impedance. The recycled, non-woven fabric/graphene ECG electrode shows good ECG signal monitoring performance and is a low-cost and environmentally friendly way to produce graphene electrodes.

Moreover, the laser-induced/scribed technology introduced above is also promising for low-cost, large-scale manufacturing of graphene electrodes [62,64,65,67]. Compared with traditional electrode fabrication methods, it has the advantage of realizing graphene fabrication and electrode patterning simultaneously.

### 3.3. Better Electrode Fitment

According to the progress of graphene ECG electrode mentioned above in recent years, much research has improved the contact between electrode and skin, such as making the electrode stick to skin more closely to reduce contact impedance, thus improving the quality of the collected signal, or using electrodes more consistent with skin surface characteristics to improve long-term comfort. In line with this trend, removing the substrate of traditional graphene electrodes and using the conductive graphene layer directly is a good choice to achieve conformable adhesion to the skin surface [26]. As shown in Figure 6e, through a sacrificial layer process, a layer of laser-scribed graphene was transferred to human skin directly with the aid of vapor. The substrate-free graphene electrode (at 2.3 mg—300 times lighter than the commercial gel electrode) is so thin and breathable that long-term wear does not lead to the accumulation of sweat, and the body feels almost nothing. Besides, the reduction in noise by removing the substrate makes the graphene electrode more sensitive to ECG signals. Apart from this, to ensure the durability of the electrode on parts of the human body with a wide range of motion, we use electrospinning fibers to strengthen the electrode. By spraying a graphene suspension, we achieved a custom electrode pattern on a breathable electrospinning fiber substrate (Figure 6f). In addition, the gecko-inspired electrode with high skin adhesion [36], the ultra-high skin-conforming graphene electrode [56], and the ultra-lightweight graphene sponge electrode [75] mentioned above are all typical methods to make the electrode better fit human skin.

### 3.4. Smart Graphene ECG Electrodes

For graphene ECG electrodes used in daily long-term ECG monitoring, apart from achieving good electrode performance and comfort, if the electrode can be endowed with diagnostic ability, the ECG system may play a more important role in daily cardiovascular health monitoring and greatly relieve medical burden. Towards this direction, an intelligent graphene nanomesh ECG electrode was developed [18,26]. In addition to the high comfort of the imperceptible graphene nanomesh ECG electrode, the supporting wearable ECG system can diagnose the acquired ECG signals by a dedicated convolutional neural network

(CNN). The algorithm can intelligently classify and diagnose four typical arrhythmias in the ECG signals (Figure 6g). The intelligent development of ECG electrodes will benefit the long-term application of graphene electrodes in daily life and disease early warning.

Moreover, not only are graphene electrodes becoming intelligent, but, in recent years, intelligent wearable health monitoring systems have become a hot research topic [89,90].

### 3.5. The Integration of Graphene ECG Electrodes

Daily health monitoring is a kind of comprehensive physiological information monitoring, which not only requires the integration of ECG electrodes into daily wear, but also requires integrated sensing, transmission, and processing of various physiological information. Multifunctional clothing is a well-integrated carrier for daily health monitoring [91,92]. In 2021, graphene-based smart clothing integrating four functions, including ECG monitoring, pressure sensing, strain sensing, and sound emitting was proposed [93]. The graphene electrodes and sensors in the clothing can detect ECG signals, breath, movement, and heart rate, which can comprehensively diagnose a person's physical condition, and, when there are health problems, the garment will start sound an alarm with an incorporated graphene acoustic device (Figure 6h). Through the integration of functions, graphene electrodes are not limited to monitoring signals, but become part of a complete daily health monitoring system. Moreover, the nylon-textile-based electrode mentioned in the fabric-based electrodes section above is also a typical case of graphene ECG electrodes integrated into garments [53].

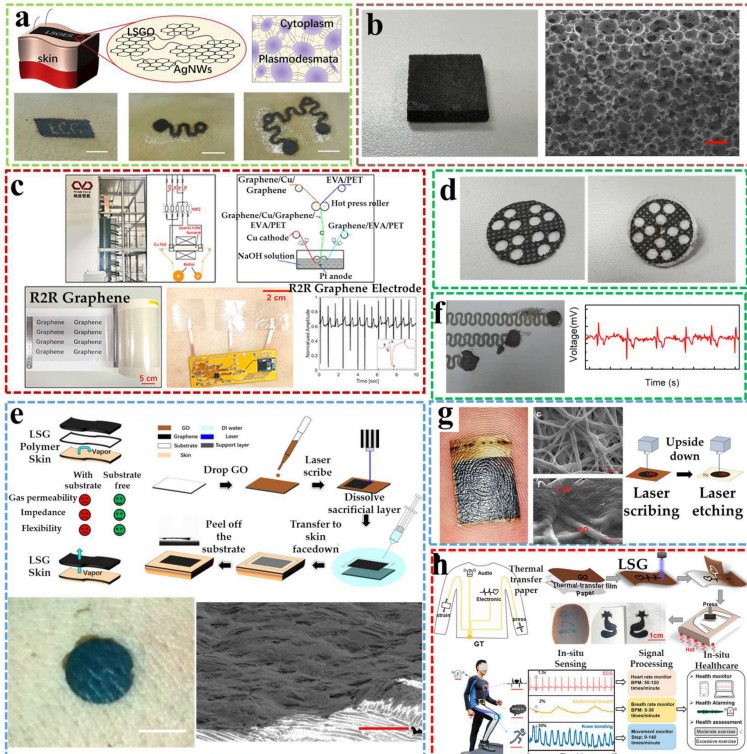

**Figure 6.** (**a**) Electrode based on silver nanowire bridging graphene. Reprinted with permission from Ref. [89]. Copyright 2020, Elsevier. (**b**) Styrene–butadiene foam/graphene ECG electrode (**left**) and its SEM image (**right**), scale bar: 300 μm. (**c**) The production of roll-to-roll graphene films and their application in ECG electrodes. Reprinted with permission from Ref. [15]. Copyright 2021, IOPscience. (**d**) The recycled non-woven fabric/graphene ECG electrode. (**e**) The fabrication process, photo, and SEM image of the substrate-free graphene electrode; scale bar: 200 μm. Reprinted with permission from Ref. [26]. Copyright 2020, American Chemical Society. (**f**) The spray-coated nanofiber-based graphene ECG electrode and acquired ECG signal. (**g**) The intelligent graphene nanomesh electrode. Reprinted with permission from Ref. [18]. Copyright 2022, Wiley. (**h**) The graphene-based multifunctional clothing for daily healthcare. Reprinted with permission from Ref. [93]. Copyright 2021, American Chemical Society.

## 4. Conclusions

Facing the need for daily long-term cardiovascular health monitoring, many advanced flexible ECG electrodes have been developed. In this review, we introduce the recent research progress of typical graphene-based ECG electrodes. Their design ideas, structural characteristics, preparation methods, and material properties are analyzed and discussed in detail, and some possible development directions of graphene ECG electrodes are proposed through our work. With the continuous development of graphene ECG electrodes by many researchers, it is believed that graphene ECG electrodes with better electrical conductivity, higher comfort, and better skin adhesion will be proposed, and more intelligent ECG monitoring systems suitable for long-term daily health monitoring will be developed soon. Relevant research and results will play an important role in actively dealing with cardiovascular health problems and an aging society.

**Author Contributions:** Conceptualization, H.T., Y.Y. and T.-L.R.; methodology, T.-R.C.; software, X.-R.H., A.-Z.Y. and Z.-K.C.; validation, J.-D.X., Y.D., W.-C.S. and Z.-Y.T.; resources, H.T.; writing—original draft preparation, T.-R.C. and H.T.; writing—review and editing, Y.Y. and T.-L.R.; visualization, T.-R.C., D.L., Y.-Z.G. and Y.W.; supervision, T.-L.R.; project administration, H.T., Y.Y. and T.-L.R.; funding acquisition, H.T., Y.Y. and T.-L.R. All authors have read and agreed to the published version of the manuscript.

**Funding:** This work was supported by the National Natural Science Foundation of China (grant nos. 62022047, 61874065, U20A20168, and 51861145202), the National Key R&D Program (grant no. 2021YFC3002200 and 2020YFA0709800), The Beijing Natural Science Foundation (grant no. M22020), the Fok Ying-Tong Education Foundation (grant no. 171051), the Beijing National Research Center for Information Science and Technology Youth Innovation Fund (grant no. BNR2021RC01007), State Key Laboratory of New Ceramic and Fine Processing of Tsinghua University (grant no. KF202109), and the Research Fund from Beijing Innovation Center for Future Chip, Center for Flexible Electronics Technology of Tsinghua University, Tsinghua-Foshan Innovation Special Fund (TFISF) (2021THFS0217) and the Independent Research Program of Tsinghua University (grant no. 20193080047).

**Institutional Review Board Statement:** Not applicable.

**Informed Consent Statement:** Not applicable.

**Data Availability Statement:** Not applicable.

**Conflicts of Interest:** The authors declare no conflict of interest.

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
