# Peer review of "Graphene-Based Flexible Electrode for Electrocardiogram Signal Monitoring"

_applsci, doi:10.3390/app12094526_

Round 1

Reviewer 1 Report

1)I suggest to add a brief part where some literature on graphene applied to biomedical devices in general is cited (1 or 2 sentences are enough). For example https://doi.org/10.1016/j.colsurfb.2019.110596 and 10.2174/1573413716666210106101124.

2) a brief part explaining how graphene electrodes work into an electrocardiogram signal monitoring device should be added

Reviewer 2 Report

The authors do a nice job of reviewing the current literature on graphene-based electrodes.  There are some minor grammatical errors that should be corrected before publication, but in general the content is fine.

Reviewer 3 Report

The authors provided a comprehensive review on the existing graphene-based flexile electrode materials for electrocardiogram. The electrode materials were categorized systematically, and their significance, fabrication methods, and highlights of properties were concisely and clearly presented. The notable research achievements were included and compared to illustrate a comprehensive picture of this field. The authors also provided several aspects that the electrode materials can be improved potentially, inspiring the peer researchers for the future study.

I agree to accept this review as presented.

Reviewer 4 Report

The captions in the picture are rather small, which makes them less readable. It would be better if you could edit that part. Other than that, I think it's a nice review paper.
